# Effect of Aspect Ratio of Ferroelectric Nanofilms on Polarization Vortex Stability under Uniaxial Tension or Compression

**DOI:** 10.3390/ma16247699

**Published:** 2023-12-18

**Authors:** Wenkai Jiang, Sen Wang, Xinhua Yang, Junsheng Yang

**Affiliations:** 1School of Mechanical Engineering, Wuhan Polytechnic University, Wuhan 430048, China; jiangwenk@whpu.edu.cn (W.J.);; 2School of Transportation, Civil Engineering and Architecture, Foshan University, Foshan 528200, China; yangxinh@hust.edu.cn

**Keywords:** stability, polarization vortex, phase field simulations

## Abstract

Mastering the variations in the stability of a polarization vortex is fundamental for the development of ferroelectric devices based on polarization vortex domain structures. Some phase field simulations were conducted on PbTiO_3_ nanofilms with an initial polarization vortex under uniaxial tension or compression to investigate the conditions of vortex instability and the effects of aspect ratio of nanofilms and temperature on them. The instability of a polarization vortex is strongly dependent on aspect ratio and temperature. The critical compressive stress increases with decreasing aspect ratio under the action of compressive stress. However, the critical tensile stress first decreases and then increases with decreasing aspect ratio, then continues to decrease. There are two inflection points in the curve. In addition, an elevated temperature makes both the critical tensile and compressive stresses decline, and will also cause the aspect ratio corresponding to the inflection point to decrease. These are very important for the design of promising nano-ferroelectric devices based on polarization vortices to improve their performance while maintaining storage density.

## 1. Introduction

In recent years, it has been discovered in some special artificial thin films and low-dimensional ferroelectric nanostructures that electric dipoles can continuously rotate to form different types of polarization vortex domain structures [1,2,3,4,5,6]. These polarization vortex structures are expected to become important candidates for high-density data storages and low-power microelectronic devices in the post-Moore’s Law period due to their novel characteristics, such as rotational chirality, strange negative capacitance, and terahertz band modulation characteristics [4,7,8,9,10]. Understanding the dynamics of polarization vortex stability is essential for the design and fabrication of ferroelectric devices founded upon polarization vortex domain structures. Vortex stability exerts a profound impact on the functional realization of devices [11,12,13,14,15]. The comprehension and exploitation of polarization vortex stability are critical to the maintenance of a stable, high-performance state in ferroelectric devices within applications by judiciously controlling environmental parameters [16,17,18,19,20,21].

A large amount of research literature indicates that ferroelectric nanofilms serving as carriers of polarization vortices play an important role in the formation and evolution of these vortices [12,22,23]. For example, Lich et al. [12] changed the shape of nanodots by adding notches or reverse notches. They used a phase field model to investigate the evolution of polarization vortices in regular ferroelectric nanodots and notched nanodots under the influence of a uniform electric field. The results indicate that ferroelectric nanodot structures with notches can induce deterministic flipping of polarization vortices by controlling the distribution of depolarization fields. By contrast, the polarization vortex in an ordinary ferroelectric nanodot structure cannot flip deterministically. Chen et al. [24] constructed a ferroelectric nanofilm-dot system by introducing epitaxially grown ferroelectric nanodots into the nanofilm, changing the geometric symmetry of the nanofilm. Phase field simulation was further used to study the feasibility of polarization vortex flipping in a system under the action of a uniform electric field. It was discovered that under suitable external conditions, a polarization single domain structure formed within the nanoscale film, while a polarization vortex domain structure emerged within the nanodots. If a uniform electric field is applied to the nanofilm, the polarization single domain in the nanofilm region can be reversed, thereby achieving a deterministic flip of the polarization vortex domain in the nanodot region. Yuan et al. [11] used phase field simulation to apply stress loading and unloading to ferroelectric nanofilms with holes. The simulation found that during the stress-induced vortex multiplication process, new vortices will be formed preferentially around the holes, while polarization vortices far away from the holes are preferentially annihilated during the vortex annihilation process. This mechanism thus enabled deterministic flipping of polarization vortices under mechanical loading. In addition, the effect of the aspect ratio of the ferroelectric nanofilms (changing the length and width of the nanofilm while keeping the area of the entire model unchanged) on the flipping of the polarization vortex during stress loading/unloading was also considered in their study. The results show that the aspect ratio of ferroelectric nanofilms has an impact on the number of vortices and the nucleation position of vortices during the evolution process, thereby affecting the flipping of polarization vortices.

The phase-field method, based on the macroscopic Ginzburg-Landau theory, employs continuously varying order parameters to formulate a system’s free energy function. This function is then used to establish control equations that describe the transient evolution of the order parameter. By introducing continuous order parameters at interfaces, the method effectively circumvents theoretical challenges associated with abrupt interface transitions, providing a unique advantage in studying the complex evolution of material microstructures. Currently, the phase-field method is widely applied in research areas such as the evolution of microstructures in ferroelectric materials and the reversal of polarization within domains influenced by external fields [25,26,27]. As a carrier for the formation of polarization vortex, the shape of ferroelectric nanofilm will directly affect the shape of polarization vortex. The impact of deterministic changes in vortex shape on their stability is not yet clear, and understanding these effects is of paramount importance for the design and optimization of new ferroelectric devices. In addition, keeping the area of the nanofilm constant and changing the aspect ratio of the film to influence the shape of the vortices is of great significance for new ferroelectric memory devices based on polarization vortices to maintain storage density while improving their performance.

In this study, the phase-field method is applied to simulate PbTiO_3_ nanofilm with initial polarization vortices. The effect of the aspect ratio of nanofilms on the stability of single vortex polarization structures under the action of uniaxial tensile or compressive stress was studied. This helps to improve the understanding and control level of polarization vortices, and is of great significance to expanding the scope of practical applications of ferroelectric nanodevices based on polarization vortex structures.

## 2. Materials and Methods

### 2.1. Phase Field Theory

In a phase field simulation of a ferroelectric system, the spontaneous polarization vector **P** = [P1, P2, P3] is commonly employed as the order parameter for characterizing polarization domain structures. The total free energy, denoted as *F*, is calculated by integrating the total free energy density (*f*) across the entire volume (*V*) of the system. According to the phenomenological theory, the total free energy can be expressed as follows [28]:(1)$$F=\int_{V}fdV=\int_{V}\left(f_{\mathrm{land}}+f_{\mathrm{grad}}+f_{\mathrm{elas}}+f_{\mathrm{coup}}+f_{\mathrm{elec}}\right)dV$$
where $f_{\mathrm{land}}$, $f_{\mathrm{grad}}$, $f_{\mathrm{elas}}$, $f_{\mathrm{coup}}$ and $f_{\mathrm{elec}}$ represent the Landau free energy density, gradient energy density, elastic energy density, coupling energy density, and electrostatic energy density, respectively.

For perovskite ferroelectrics, the Landau energy density $f_{\mathrm{land}}$ is often expressed as a sixth-order polynomial of the spontaneous polarization components, as represented by [28,29,30]:(2)$$\begin{array}{l} f_{\mathrm{land}}=\alpha_{1}\left(P_{1}^{2}+P_{2}^{2}+P_{3}^{2}\right)+\alpha_{11}\left(P_{1}^{4}+P_{2}^{4}+P_{3}^{4}\right)\\ \;+\alpha_{12}\left(P_{1}^{2}P_{2}^{2}+P_{2}^{2}P_{3}^{2}+P_{1}^{2}P_{3}^{2}\right)+\alpha_{111}\left(P_{1}^{6}+P_{2}^{6}+P_{3}^{6}\right)\\ \;+\alpha_{112}\left[P_{1}^{4}\left(P_{2}^{2}+P_{3}^{2}\right)+P_{2}^{4}\left(P_{1}^{2}+P_{3}^{2}\right)+P_{3}^{4}\left(P_{1}^{2}+P_{2}^{2}\right)\right]+\alpha_{123}P_{1}^{2}P_{2}^{2}P_{3}^{2} \end{array}$$

Here, α1, α11, α12, α111, α112, and α123 represent phenomenological coefficients. The coefficient α1 exhibits a linear temperature dependence according to the Curie-Weiss law, which can be expressed as: $\alpha_{1}=\left(T-T_{0}\right)/2k_{0}c_{0}$, where *T* and *T*_0_ denote the temperature and the Curie-Weiss temperature, respectively, while *k*_0_ represents the dielectric constant of a vacuum, and *c*_0_ is the Curie constant.

To account for the spatially nonuniform distribution of polarization within the system, it becomes necessary to incorporate the contribution of gradient energy into the total free energy. In the lowest order of Taylor expansion, the gradient energy density [31,32,33] $f_{\mathrm{grad}}$ can be described as follows:(3)$$\begin{array}{l} f_{\mathrm{grad}}=\frac{1}{2}G_{11}\left(P_{1,1}^{2}+P_{2,2}^{2}+P_{3,3}^{2}\right)+G_{12}\left(P_{1,1}P_{2,2}+P_{2,2}P_{3,3}+P_{1,1}P_{3,3}\right)\\ \;\;+\frac{1}{2}G_{44}\left[{\left(P_{1,2}+P_{2,1}\right)}^{2}+{\left(P_{1,3}+P_{3,1}\right)}^{2}+{\left(P_{2,3}+P_{3,2}\right)}^{2}\right]\\ \;\;+\frac{1}{2}G_{44}^{\prime }\left[{\left(P_{1,2}-P_{2,1}\right)}^{2}+{\left(P_{1,3}-P_{3,1}\right)}^{2}+{\left(P_{2,3}-P_{3,2}\right)}^{2}\right] \end{array}$$
where $G_{11}$, $G_{12}$, $G_{44}$ and $G_{44}^{\prime }$ are the gradient energy coefficients, and *P_i,j_* with *i*, *j* = 1, 2, and 3 denotes the partial derivative of the *I* component of the polarization vector with respect to the *j* coordinate.

The elastic energy is induced by mechanical strain, and its density $f_{\mathrm{elec}}$ can be written as follows:(4)$$f_{\mathrm{elas}}=\frac{1}{2}c_{11}\left(\varepsilon_{11}^{2}+\varepsilon_{22}^{2}+\varepsilon_{33}^{2}\right)+c_{12}\left(\varepsilon_{11}\varepsilon_{22}+\varepsilon_{22}\varepsilon_{33}+\varepsilon_{11}\varepsilon_{33}\right)+2c_{44}\left(\varepsilon_{12}^{2}+\varepsilon_{23}^{2}+\varepsilon_{13}^{2}\right)$$
where $c_{11}$, $c_{12}$ and $c_{44}$ are the elastic constants and $\varepsilon_{ij}$ is the elastic strain component.

The coupling energy density between spontaneous polarization and strain is given by $f_{\mathrm{coup}}$ and can be expressed as follows [28]:(5)$$\begin{array}{l} f_{\mathrm{coup}}=-q_{12}\left[\varepsilon_{11}\left(P_{3}^{2}+P_{2}^{2}\right)+\varepsilon_{22}\left(P_{1}^{2}+P_{3}^{2}\right)+\varepsilon_{33}\left(P_{1}^{2}+P_{2}^{2}\right)\right]\\ \;\;-2q_{44}\left(\varepsilon_{12}P_{1}P_{2}+\varepsilon_{13}P_{1}P_{3}+\varepsilon_{23}P_{2}P_{3}\right)-q_{11}\left(\varepsilon_{11}P_{1}^{2}+\varepsilon_{22}P_{2}^{2}+\varepsilon_{33}P_{3}^{2}\right) \end{array}$$
where $q_{11}$, $q_{12}$ and $q_{44}$ are the electrostriction coefficients.

Furthermore, electrostatic energy arises from the Coulomb interactions between electric dipoles, and its density is denoted as $f_{\mathrm{elec}}$. It can be expressed as follows [34,35]:(6)$$f_{\mathrm{elec}}=-\frac{1}{2}k_{c}\left(E_{1}^{2}+E_{2}^{2}+E_{3}^{2}\right)-E_{1}P_{1}-E_{2}P_{2}-E_{3}P_{3}$$
where $k_{c}$ is the dielectric constant of the background material.

The time-dependent evolution of the polarization field toward its thermodynamic equilibrium state can be characterized by the following time-dependent Ginzburg-Landau equation:(7)$$\frac{\partial P_{i}}{\partial t}=-L\frac{\delta F}{\delta P_{i}}(i=1,\;2,\;3)$$
where *L* represents a kinetic coefficient associated with domain wall mobility, *t* is the time, and $\delta F/\delta P_{i}$ denotes the thermodynamic driving force governing the spatial and temporal evolution of polarization.

Furthermore, for a system with free body force and free charge, it is essential that the following mechanical equilibrium equation and Maxwell’s equations be simultaneously satisfied.
(8)$$\frac{\partial }{\partial x_{j}}\left(\frac{\partial F}{\partial \varepsilon_{ij}}\right)=0$$
and
(9)$$\frac{\partial }{\partial x_{i}}\left(-\frac{\partial F}{\partial E_{i}}\right)=0$$
in which the stress and electric displacement can be derived from $\sigma_{ij}=\partial F/\partial \varepsilon_{ij}$ and $D_{i}=-\partial F/\partial E_{i}$, respectively. A numerical algorithm based on the finite element method [20] can be developed to solve Equations (7)–(9).

### 2.2. Numerical Model

The shape and size of our PbTiO_3_ nanofilm are shown in Figure 1, $l_{a}=a\;\mathrm{nm}$, $l_{b}=b\;\mathrm{nm}$. We first establish a Cartesian coordinate system with its origin centered on the nanofilm. The X_1_ axis aligns with the nanofilm’s length, while the X_2_ axis aligns with its width. The model’s mechanical boundary condition dictates that there is zero stress on the surface, expressed mathematically as $\;\sigma_{ij}n_{j}=0$. Additionally, the electrical boundary condition specifies an open circuit, mathematically represented as $\;D_{i}n_{i}=0$ [36,37]. A two-dimensional (2D) discrete grid is employed, with a minimum unit size smaller than 0.4 nm. Each node’s initial value, which represents the magnitude and direction of the polarization vector, is assigned randomly and is significantly smaller in magnitude, being two orders of magnitude lower than the spontaneous polarization. The next step is to simulate the evolution of the polarization structure over time until a stable polarization state is obtained. The material parameters are from reference [19,38,39,40]. The vortex domain structure is characterized by applying the curl operator (∇ ×) to the polarization vector M, denoted as ∇ × P [1]. It is observed that a spontaneous vortex naturally forms at the center of a nanofilm.

Subsequently, keeping the area of the nanofilm constant, $l_{a}l_{b}$ = 100 nm^2^, we modify the film’s geometry by altering the aspect ratio, represented as $\eta =l_{a}/l_{b}$. The stable polarization vortex structure formed in PbTiO_3_ nanofilms of different shapes is used as the initial polarization. At a given temperature, increasing tensile stress or compressive stress is applied in the X_1_-axis direction to the nanofilm with the initial polarization until the vortex loses stability. A schematic diagram of mechanical stress application is shown in Figure 2. Throughout the calculation process, the initial temperature conditions are maintained, and the stress increases in equal increments from zero. Each increment is set at Δσ = 0.01 GPa. Each loading step is maintained long enough to allow the polarization structure to stabilize. When the stress increases to a critical value, the polarization vortex will lose stability. The critical tensile stress and critical compressive stress are recorded as σt and σp, respectively. In order to evaluate the effects of the nanofilm’s shape and temperature on the evolution and stability of polarization vortices, the aspect ratio *η* ranges from 0.8 to 1, with one selected every 0.02, for a total of 11. The temperature *T* is selected by considering a range from 100 K to 500 K, and one is selected every 100 K. A total of 5 temperature conditions are selected. The results obtained from computations involving these various combinations will undergo thorough analysis and discussion in the subsequent sections. The vortex shape ratio *R* [19] is used to characterize the shape of the polarization vortex. The shape ratio *R* of the initial polarization vortex is affected by the aspect ratio *η* of the nanofilm.

## 3. Results

### 3.1. Instability of Polarization Vortices under the Action of Compressive Stress

First, the instability process of polarization vortices under the action of compressive stress was simulated. Figure 3 shows the evolution of the polarization structure with time when a PbTiO_3_ nanofilm with an aspect ratio *η* = 0.88 is subjected to a critical compressive stress of −4.36 GPa at 300 K. It can be seen from the figure that the instability of the polarization vortex is an α-type instability [19].

As can be seen in Figure 3a, due to the change in the aspect ratio *η* of the nanofilm, the shape of the vortex in the initial structure also changed significantly. The vortex shape ratio *R* is introduced as a parameter to characterize the polarization vortex shape [19]. In Figure 3a, the initial polarization vortex shape ratio is *R* = 1.54. From Figure 3a,b, the number and direction of vortices have not changed, but the shape of the vortices has changed, and the vortices are in the deformation stage. In Figure 3c,d, the number of vortices changes, the initial polarization vortex loses stability. The vortex is in the instability stage. At this stage, the initial vortex is annihilated, and a new vortex is formed and gradually stabilizes. When the polarization structure is finally stabilized, only two vortices remain in the nanofilm, which have a “counterclockwise-clockwise” chirality from left to right. The shape of the initial vortex changes with the change of the aspect ratio of the nanofilm, and the long axis of the vortex is along the X_2_ direction. Under the action of compressive stress, the initial polarization vortex loses its stability, and the re-formed new vortex is arranged along the X_1_ direction, with its long axis parallel to the X_2_ axis, consistent with the long axis of the initial vortex.

Subsequently, the effects of temperature and film aspect ratio on polarization vortex instability were investigated. The simulation found that within the studied temperature and film aspect ratio ranges, the instability process of the vortex was consistent with the instability process shown in Figure 3. Under the action of compressive stress, changes in the temperature and film aspect ratio do not change the instability type of the vortex. Figure 4 shows the variation curve of the critical compressive stress *σ*_p_ with the aspect ratio *η* of the nanofilm at different temperatures *T*. It can be seen that at different temperatures, the critical compressive stress has the same changing trend with the change of the aspect ratio of the nanofilm. As the aspect ratio of the nanofilm increases, the critical compressive stress has an obvious and approximately linear downward trend. If the nanofilm aspect ratio remains constant, the critical compressive stress decreases with increasing temperature. Therefore, under conditions where the aspect ratio of the nanofilm is large or the temperature is high, the vortex will be more likely to lose stability. The calculation results show that changing the aspect ratio of the nanofilm can adjust the resistance of polarization vortices to compressive stress.

### 3.2. Instability of Polarization Vortices under the Action of Tensile Stress

Figure 5 shows the evolution of the polarization structure with time when a PbTiO_3_ nanofilm with an aspect ratio *η* = 0.88 is subjected to a critical tensile stress of 6.83 GPa at 300 K. In Figure 5a–c, the shape of the vortex changes, but the number and direction of rotation remain unchanged, and the vortex is in the deformation stage. In Figure 5a, the vortex shape ratio *R* of the initial structure is greater than 1, while in Figure 5c the vortex shape ratio R is less than 1. At this stage, the vortex shape ratio gradually decreases from *R* greater than 1 to *R* less than 1. The long axis corresponding to the polarization vortex changes from the X_2_ direction to the X_1_ direction. Finally, the original vortex in the nanofilm was annihilated, and three new vortices were generated. They have a “clockwise-counterclockwise-clockwise” vortex chirality from top to bottom. The nanofilm serves as the carrier of polarization vortices. The change in the aspect ratio of the film directly affects the shape of the initial vortex. The long axis of the vortex is in the X_2_ axis direction. The initial polarization vortex loses stability under the action of tensile stress. After instability, the number of vortices changes, the new vortices formed are arranged along the X_2_ axis, and their long axes are parallel to the X_1_ axis. This shows that the instability of polarization vortices under the action of tensile stress will be accompanied by changes in the number and axis direction of the vortices.

Figure 6 presents the variation curve of the critical tensile stress σ_t_ with the aspect ratio *η* of the nanofilm at different temperatures *T*. Apart from the critical compressive stress, Figure 6 reveals a distinct pattern in the behavior of critical tensile stress as the aspect ratio of the nanofilm gradually transitions from 1 to 0.8. This transition can be categorized into three distinct stages: an initial decrease, followed by an increase, and ultimately a further decrease. Overall, as the aspect ratio of the nanofilm decreases, the change in critical tensile stress shows a decreasing trend. An important observation is the presence of a critical tensile stress minimum between the first and second stages, as well as a critical tensile stress maximum between the second and third stages. Of particular interest is the critical tensile stress maximum value, which captures our primary attention when compared to the critical tensile stress minimum value.

The vortex instability shown in Figure 5 is located in the second stage of the curve in Figure 6. The vortex instability evolution processes in the first and third stages of the curve are shown in Figure 5 and Figure 6, respectively. Figure 7 depicts the temporal evolution of the polarization structure for a nanofilm with an aspect ratio *η* of 0.98 subjected to a critical tensile stress of 7.31 GPa at 300 K. In this figure, the polarization vortex’s instability process resembles the scenario presented in Figure 5. In Figure 7a–c, the vortex undergoes a deformation phase, during which the vortex’s shape ratio, denoted as *R*, gradually transitions from being greater than 1 to less than 1. Additionally, the long axis of the vortex also changes from the X_2_ axis direction to the X_1_ axis direction. The vortex instability observed in the first phase of the curve is akin to that in the second phase. However, due to the influence of the nanofilm’s aspect ratio, the initial vortex shape ratio in the second phase is greater than that in the first phase.

Figure 8 shows the evolution of the polarization structure with time when a PbTiO_3_ nanofilm with an aspect ratio *η* = 0.84 is subjected to a critical tensile stress of 5.29 GPa at 300 K. The instability form of the vortex is very different from that of the first and second stages. In Figure 8a,b, the vortices remain unchanged in number and rotation direction, but the shape changes, and the vortex is in the deformation stage. In this stage, the vortex shape ratio changes continuously, *R* gradually increases from 1.89 to 3.40, and the direction of the long axis of the vortex remains unchanged. When the time step *t* = 189 ns, as shown in Figure 8c, an anti-vortex structure is formed in the middle of the vortex, dividing the initial vortex into two. At this time, the number of vortices changes and the vortex enters the instability stage. When *t* = 194 ns, as shown in Figure 8e, the anti-vortex structure gradually evolves into a 180° domain wall, becoming the interface between the two vortices. As the polarization configuration stabilizes, the 180° domain wall evolves into a new vortex. Three vortices are generated in the stabilized nanofilm, and they have a “clockwise-counterclockwise-clockwise” vortex chirality from top to bottom. The simulation results show that when the film aspect ratio decreases to a certain extent, the curve enters the third stage. Compared with the first two stages, although the vortex instability type is still α-type instability, there are obvious differences in the vortex instability process whether in the deformation stage or the instability stage. In the deformation stage, the vortex shape ratio does not decrease but increases, while in the instability stage, an anti-vortex configuration is formed in the middle of the vortex, dividing the initial vortex into two. The type of vortex instability that occurs in the third stage is called α’ type instability.

Comparing Figure 4 and Figure 6, when the critical tensile stress value of the polarization vortex is relatively large, the critical compressive stress value is generally small. Similarly, when the critical compressive stress value of the polarization vortex is relatively large, the critical tensile stress value will also be relatively small. Within the studied aspect ratio range, the film aspect ratio corresponding to the maximum critical tensile stress, the maximum critical compressive stress, and the maximum value of the critical tensile stress is worth noting. Taking the temperature *T* = 300 K as an example, a comparative analysis of the critical tensile stress and critical compressive stress corresponding to the aspect ratios of these three locations was conducted. The nanofilm aspect ratios corresponding to the maximum critical tensile stress, the maximum critical compressive stress and the maximum critical tensile stress are *R* = 1, *R* = 0.8 and *R* = 0.88 respectively.

Under the condition *R* = 1, the critical tensile stress and critical compressive stress are recorded at 8.10 GPa and −3.26 GPa, respectively, with the maximum critical tensile stress observed at this point. For the scenario when *R* = 0.80, the critical tensile stress and critical compressive stress are measured at 4.55 GPa and −5.54 GPa, respectively, and the maximum critical compressive stress is observed. At *R* = 0.88, the critical tensile stress and critical compressive stress are determined to be 6.82 GPa and −4.35 GPa, respectively. When compared to the *R* = 1 case, the critical tensile stress is reduced by 15.8%, while the critical compressive stress shows a 33.4% increase. As a result, it is evident that one can attain significantly enhanced overall performance by selecting an appropriate aspect ratio in accordance with specific practical requirements.

## 4. Discussion

### 4.1. Effect of Vortex Aspect Ratio on Vortex Stability

The aspect ratio of nanofilms has an important influence on the shape of the initial polarization vortex. When the aspect ratio is less than 1, the shape ratio *R* of the initial vortex is greater than 1, and the long axis of the vortex is parallel to the X_2_ axis. As can be seen in Figure 3, after the polarization vortex becomes unstable under the influence of compressive stress, the long axis of the new vortex formed in the film is consistent with the long axis direction of the initial vortex. During the vortex instability process, the number of vortices changes, but there is no change in the direction of the vortex axis. In Figure 5, Figure 7, and Figure 8, after the polarization vortex becomes unstable under the action of tensile stress, the long axis of the new vortex formed in the film is perpendicular to the long axis of the initial vortex. These observations collectively indicate that not only does the number of vortices change, the direction of the vortex axis also changes during the vortex instability process. It is shown that the type of external load affects the change in the direction of the vortex axis. Under the influence of compressive stress, the smaller the aspect ratio of the film, the greater the critical compressive stress, and the stronger the vortex’s ability to resist load. Under the action of tensile stress, as the aspect ratio of the film decreases, the critical tensile stress gradually increases. Therefore, the polarization vortex loses stability under a certain mechanical load. If the long axis of the unstable vortex is parallel to the long axis of the initial vortex, the smaller the aspect ratio of the nanofilm, the greater the critical instability load of the vortex. If the long axis of the vortex after losing stability is perpendicular to the long axis of the initial vortex, the smaller the aspect ratio of the nanofilm, the smaller the critical instability load of the vortex. In their study on the influence of aspect ratio on polarization vortex flipping in ferroelectric nanofilms with holes, Yuan et al. [11] observed that changes in the nanofilm’s aspect ratio directly affected the critical load for vortex instability. Their computed results are consistent with our calculations.

Under the action of mechanical load, whether the axis direction changes after vortex instability has a significant impact on the critical load of vortex instability. The polarization vortex loses stability under a certain mechanical load. If the long axis of the new vortex formed after the destabilization is parallel to the long axis of the initial vortex, then the ability of the vortex to resist the load increases as the aspect ratio of the film decreases. If the long axis of the new vortex formed after instability is perpendicular to the long axis of the initial vortex, then the ability of the vortex to resist this load will weaken as the nanofilm aspect ratio decreases.

### 4.2. Effect of Vortex Long Axis on Vortex Stability

It can be seen from Figure 6 that as the aspect ratio of the nanofilm decreases, the change of critical tensile stress is divided into three stages. The simulation results show that vortex instability is accompanied by changes in the direction of the long axis of the vortex. The first stage of the curve in Figure 6 corresponds to Figure 7, and the second stage of the curve corresponds to Figure 5. The vortex shape ratio changes continuously from *R* greater than 1 to *R* less than 1, and the direction of the long axis of the vortex also changes. The third stage of the curve corresponds to Figure 8. Due to the formation of the anti-vortex, the long axis of the vortex undergoes a sudden change. The shape ratio of the initial vortex has a significant impact on whether the direction of the long axis of the vortex changes.

In Figure 6, the vortex instability processes corresponding to the first stage and the second stage of the curve are very similar, but the changes in the curves in these two stages are obviously different. In order to analyze this effect, the vortex instability process corresponding to the first and second stages of the curve must next be analyzed. It is observed that the long axis of the polarization vortex tends to remain in the original long axis direction. In these two stages, there is an intermediate state that exists for a long time during the configuration evolution process, as shown in Figure 5b and Figure 7b. This intermediate state is very close to the state after the vortex is stabilized under the action of near-critical tensile stress (a tensile stress that is one stress increment smaller than the critical tensile stress). This suggests that vortices tend to maintain this state before losing stability.

Figure 9 shows the initial polarization vortex when the aspect ratio of the nanofilm is *η* = 1, *η* = 0.96, *η* = 0.92, and *η* = 0.88, respectively. The corresponding vortex shape ratios are *R* = 1, *R* = 1.11, *R* = 1.28, and *R* = 1.54, respectively. A near-critical tensile stress is applied to the initial polarization vortex until a stable polarization structure is obtained, as shown in Figure 10. The shape ratios of the vortices in the figure are *R* = 0.34, *R* = 0.46, *R* = 0.67 and *R* = 1 respectively. In order to analyze the changes in the vortex shape, Figure 11 shows the variation curves of *R*1, *R*2 and Δ*R* with the aspect ratio of the nanofilm. Among them, *R*1 represents the shape ratio of the initial polarization vortex, *R*2 represents the vortex shape ratio under the action of near-critical tensile stress, and Δ*R* represents the change in the vortex shape ratio. It is evident that as the aspect ratio of the nanofilm decreases, R1 increases, and R2 increases under the influence of approaching critical tensile stress, while ΔR decreases. This indicates that the larger the initial vortex shape ratio is, the smaller the change in vortex shape is. The long axis of the vortex tends to maintain the original direction and shows resistance to changes in the direction of the long axis. When the vortex shape ratio increases to a certain value, the vortex instability type changes, and the curve in Figure 6 enters the third stage. In the first and second stages of the curve, the smaller the nanofilm aspect ratio, the larger the shape ratio of the initial polarization vortex, and the stronger the vortex’s ability to resist external loads.

The aspect ratio of the nanofilm influences the shape of the initial polarization vortex, consequently impacting the stability of the polarization vortex. Two crucial factors contribute significantly to this phenomenon. The first factor is whether the direction of the long axis of the vortex changes before and after instability. Under a specific mechanical load, vortex instability is observed. When the newly formed vortex, post-instability, maintains alignment of its long axis with that of the initial vortex, the vortex’s capacity to resist this load is enhanced as the aspect ratio of the nanofilm decreases. Conversely, if the long axis of the newly formed vortex, post-instability, becomes orthogonal to the long axis of the initial vortex, the vortex’s resistance to the load weakens as the aspect ratio of the film decreases. The second factor is the initial vortex shape ratio. Specifically, under the action of external loads, the continuous change of the shape ratio of the polarization vortex causes the direction of the long axis of the vortex to change. In this type of instability, the ability of the vortex to resist external loads increases as the initial vortex shape ratio increases.

Under the action of compressive stress, the stability of polarization vortices is mainly affected by the first factor. Therefore, as the aspect ratio of the nanofilm decreases, the critical compressive stress gradually increases, and the ability of the vortices to resist compressive stress gradually increases. Under the action of tensile stress, the stability of polarization vortices is affected by two factors at the same time. In the first stage of the curve, the initial vortex shape is relatively small, the influence of Factor Two is secondary, and the influence of Factor One is dominant. Therefore, at this stage, as the aspect ratio of the nanofilm decreases, the critical tensile stress gradually decreases, and the ability of the vortex to resist tensile stress gradually weakens. Overall, in the first and second stages of the curve, as the initial vortex shape ratio gradually increases, the effect of Factor Two gradually increases, and the change in the slope of the curve also reflects the gradual increase in the influence of Factor Two. In the third stage of the curve, the stability of the polarization vortex is only affected by the first factor. At this stage, as the aspect ratio of the nanofilm decreases, the critical compressive stress gradually decreases, and the ability of the vortices to resist tensile stress gradually weakens.

## 5. Conclusions

PbTiO_3_ nanofilms with an initial polarization vortex are modeled and their vortex instability processes under uniaxial tension or compression are simulated with the phase field theory. The following conclusions are given.

The aspect ratio of nanofilms directly shapes the initial polarization vortex, influencing its stability. The critical compressive stress rises with the vortex shape ratio, whereas the critical tensile stress initially weakens, then strengthens, and finally weakens with the shape ratio, reaching a maximum. This maximum critical tensile stress decreases with temperature elevation.The stability of polarization vortices depends on the initial orientation of the vortex’s major axis. If, during instability caused by mechanical loads, the vortex’s major axis remains parallel to the initial axis, its resistance increases with decreasing nanofilm aspect ratio. Conversely, if the vortex’s major axis becomes perpendicular post-instability, resistance decreases with a decreasing aspect ratio of the nanofilm.During the process of vortex instability, if there is a change in the long-axis direction of the vortex and this change is caused by the continuous variation of the vortex shape ratio, then in such a scenario of vortex instability, the larger the initial vortex shape ratio, the stronger the vortex’s resistance to external loads.

In summary, adjusting the aspect ratio of nanofilms reveals crucial factors influencing the stability of polarization vortices, which has significant implications for the design and performance optimization of novel ferroelectric storage devices based on polarization vortices. Within the constraint of maintaining a constant nanofilm area, altering the aspect ratio to influence the shape of polarization vortices is of paramount importance for enhancing the performance of these innovative ferroelectric storage devices while preserving storage density.

## Figures and Tables

**Figure 1 materials-16-07699-f001:**
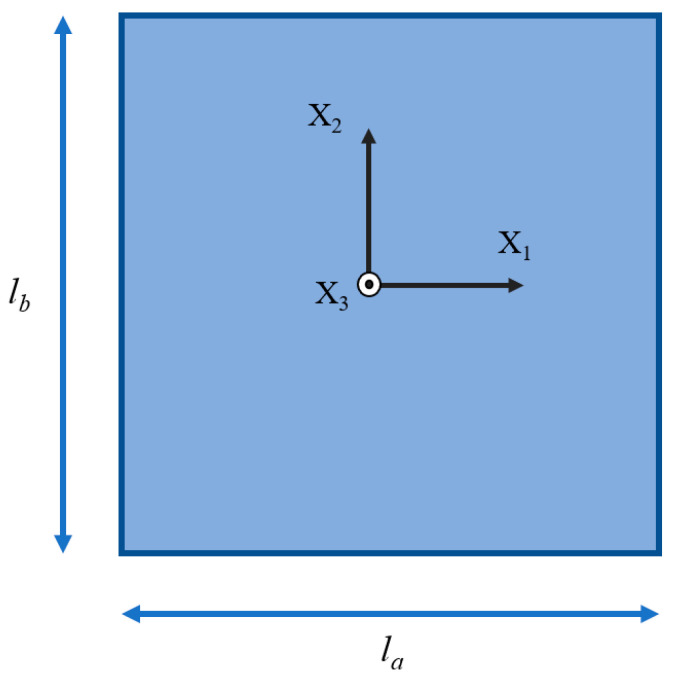
Model of PbTiO₃ nanofilm with Xi representing the coordinate axis.

**Figure 2 materials-16-07699-f002:**
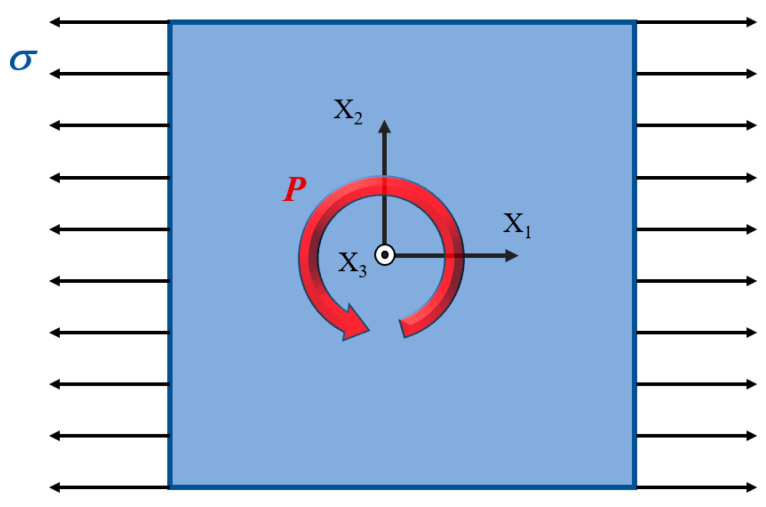
Representation of a PbTiO₃ nanofilm model subjected to linearly increasing tensile or compressive stress. P represents the polarization vector, and σ denotes stress.

**Figure 3 materials-16-07699-f003:**
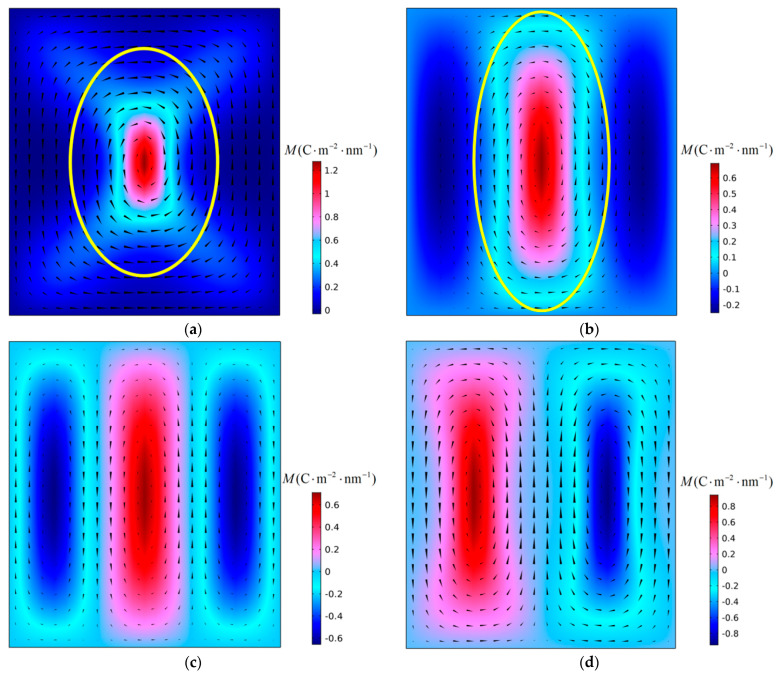
Polarization vortex configurations in the PTO nanofilm with an aspect *η* = 0.88 under compressive stress of 4.36 GPa at (**a**) *t* = 0 ns, (**b**) *t* = 142 ns, (**c**) *t* = 180 ns, and (**d**) *t* = 2000 ns. The time step is 1 ns.

**Figure 4 materials-16-07699-f004:**
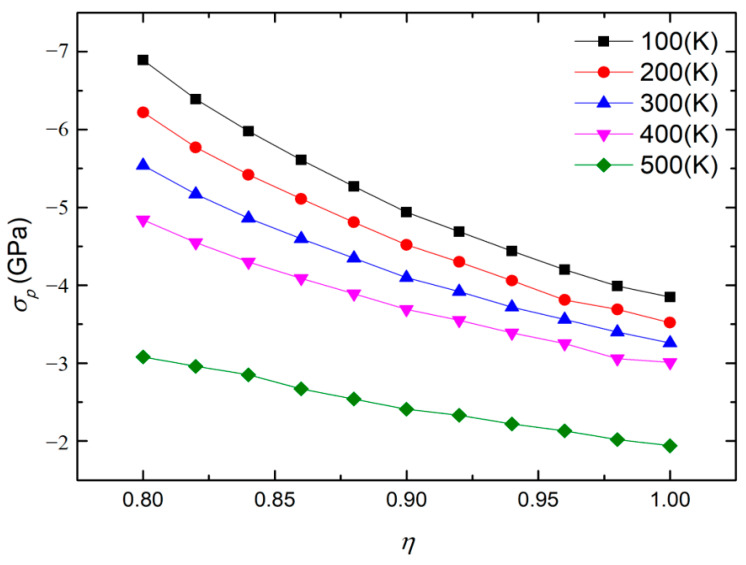
Variation of the critical compressive stress with the aspect ratio of nanofilms at different temperatures.

**Figure 5 materials-16-07699-f005:**
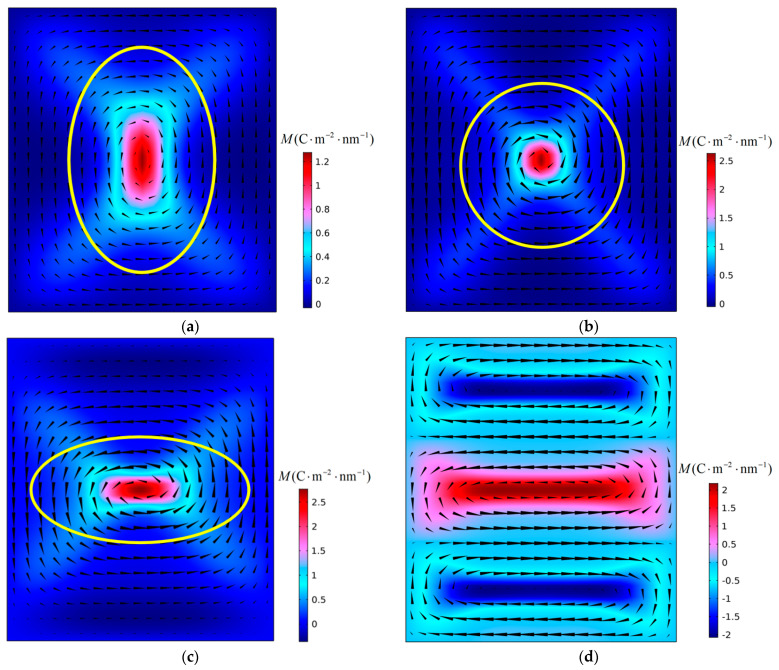
Polarization vortex configurations in the PTO nanofilm with an aspect *η* = 0.88 under tensile stress of 6.83 GPa at (**a**) *t* = 0 ns, (**b**) *t* = 75 ns, (**c**) *t* = 439 ns, and (**d**) *t* = 2000 ns.

**Figure 6 materials-16-07699-f006:**
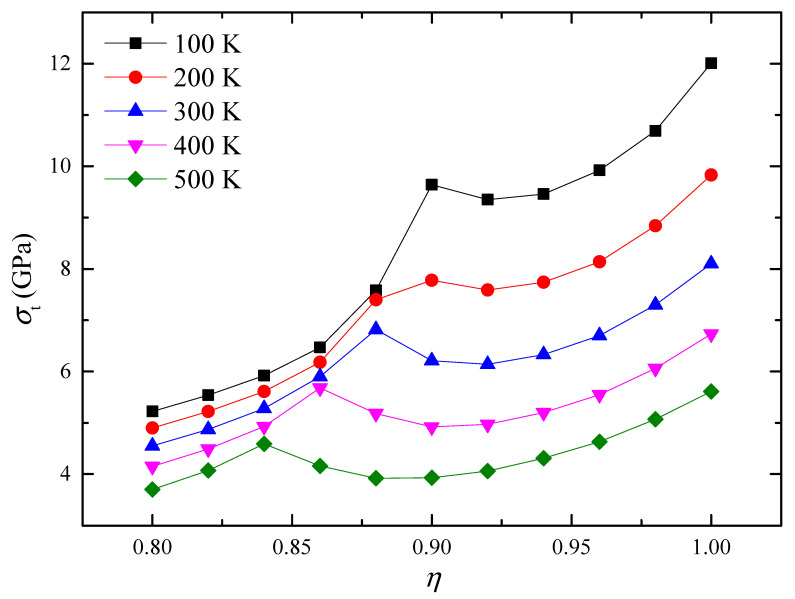
Variation of the critical tensile stress with the aspect ratio of nanofilms at different temperatures.

**Figure 7 materials-16-07699-f007:**
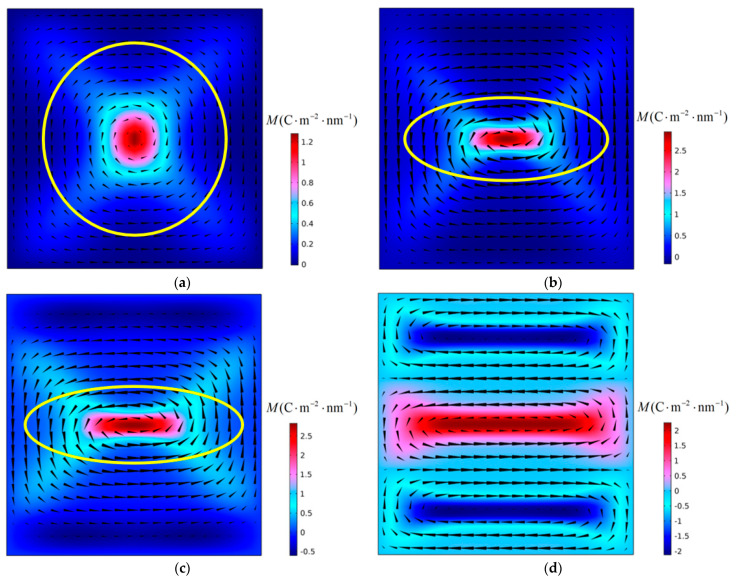
Polarization vortex configurations in the PTO nanofilm with an aspect ratio *η* = 0.98 under tensile stress of 7.31 GPa at (**a**) *t* = 0 ns, (**b**) *t* = 75 ns, (**c**) *t* = 297 ns, and (**d**) *t* = 2000 ns.

**Figure 8 materials-16-07699-f008:**
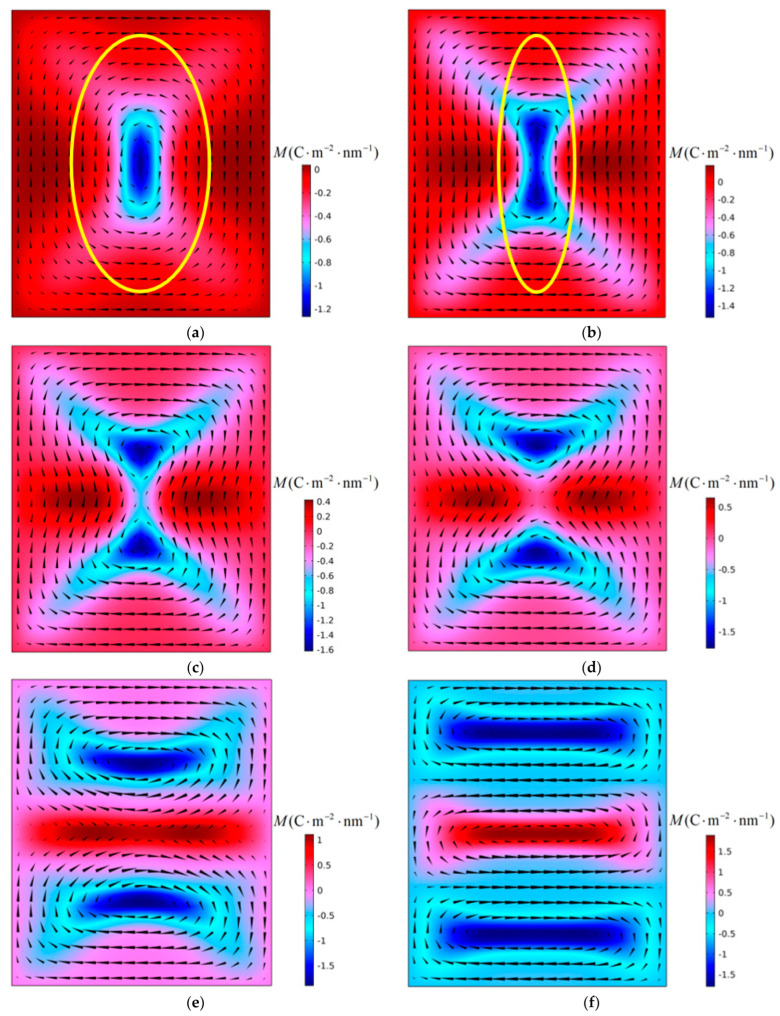
Polarization vortex configurations in the PTO nanofilm with an aspect ratio *η* = 0.84 under tensile stress of 5.29 GPa at (**a**) *t* = 0 ns, (**b**) *t* = 15 ns, (**c**) *t* = 189 ns, (**d**) *t* = 192 ns, (**e**) *t* = 194 ns, and (**f**) *t* = 2000 ns.

**Figure 9 materials-16-07699-f009:**
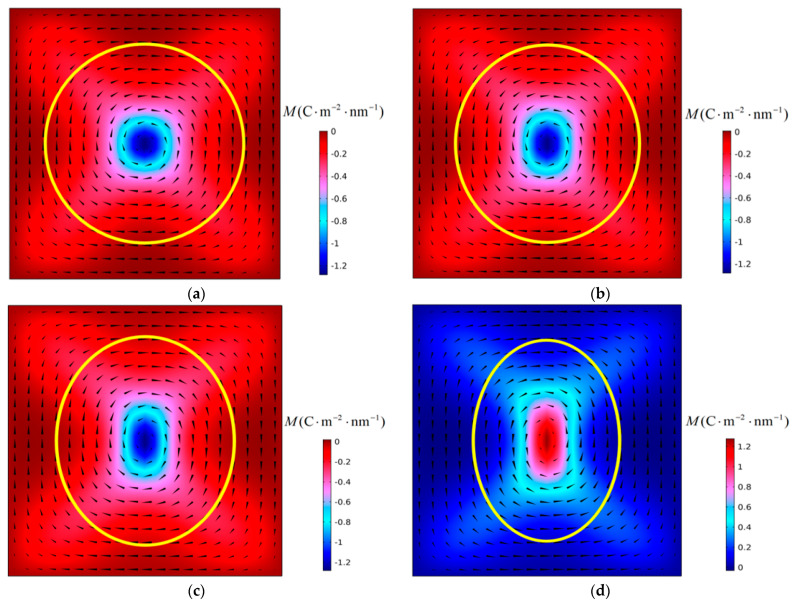
Initial polarization vortex configuration formed in nanofilms under aspect ratios (**a**) *η* = 1, (**b**) *η* = 0.96, (**c**) *η* = 0.92 and (**d**) *η* = 0.88.

**Figure 10 materials-16-07699-f010:**
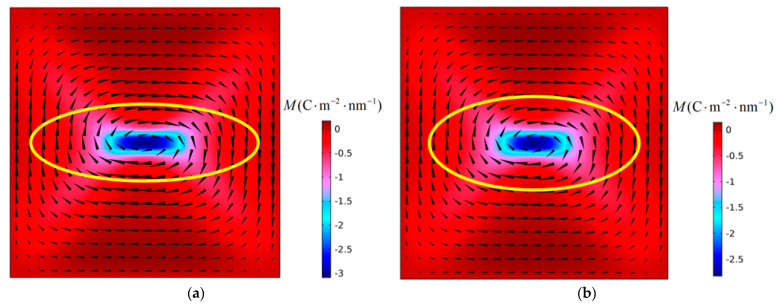
Polarization vortex configurations in nanofilms with different aspect ratios (**a**) *η* = 1, (**b**) *η* = 0.96, (**c**) *η* = 0.92 and (**d**) *η* = 0.88 under the action of near-critical tensile stress.

**Figure 11 materials-16-07699-f011:**
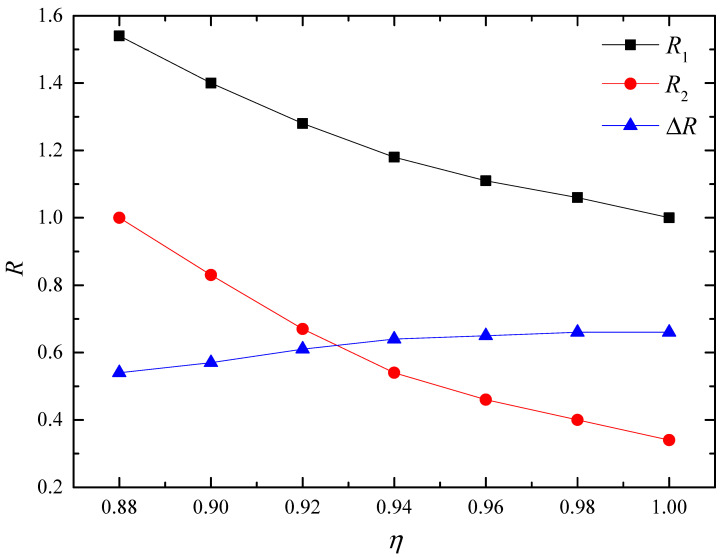
Curves of initial vortex shape ratio *R*1, near-critical tensile stress vortex shape ratio *R*2, and vortex shape ratio change Δ*R* with the aspect ratio of nanofilms.

## Data Availability

Data are contained within the article.

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
