# Peer review of "Effect of Aspect Ratio of Ferroelectric Nanofilms on Polarization Vortex Stability under Uniaxial Tension or Compression"

_materials, 2023, doi:10.3390/ma16247699_

Round 1

Reviewer 1 Report

Comments and Suggestions for Authors

 Several matters posed in the paper need to be addressed:

1)      Section 1, Introduction: The motivation/reason to conduct the phase simulation on PbTiO3 nanofilms; based on Ginzburg-Landau Theory is unclear.

2)      Section 3, Results: This section is too long and some parts deemed to be repetitive with Section 4: Discussion. It is recommended to simplify and cut down the repetitive details in Section 3.

3)      Section 4: Discussion- There are several formatting errors especially on Pg 17.

4)      Section 5: Conclusion need to be improved to highlight the objective of research work and demonstrate the importance of the paper to the field of ferroelectric nanofilms. Suggestions for further modification and improvement could be included.

5)      References- Most of the listed  references are more than 5 years. Authors are encouraged to include newer and updated references.

Comments on the Quality of English Language

This draft needs to be carefully checked for proper English expression, formatting, font size, font format and spacing errors before submitting a new revised version.

Author Response

Comments 1: Section 1, Introduction: The motivation/reason to conduct the phase simulation on PbTiO3 nanofilms; based on Ginzburg-Landau Theory is unclear.

Response: Acknowledged. We have clarified the motivation in the introduction, linking it explicitly to the Ginzburg-Landau Theory. Please refer to the revised Section 1 for clarity.

The new content is as follows: The phase-field method, based on the macroscopic Ginzburg-Landau theory, employs continuously varying order parameters to formulate a system's free energy function. This function is then used to establish control equations that describe the transient evolution of the order parameter. By introducing continuous order parameters at interfaces, the method effectively circumvents theoretical challenges associated with abrupt interface transitions, providing a unique advantage in studying the complex evolution of material microstructures. Currently, the phase-field method is widely applied in research areas such as the evolution of microstructures in ferroelectric materials and the reversal of polarization within domains influenced by external fields.

Comments 2: Section 3, Results: This section is too long and some parts deemed to be repetitive with Section 4: Discussion. It is recommended to simplify and cut down the repetitive details in Section 3.

Response: Understood. We have streamlined Section 3, removing redundancies. The revised section is more concise and focused. Refer to the updated Section 3.

Comments 3: Section 4: Discussion There are several formatting errors especially on Pg 17.

Response: Thank you for highlighting this issue. We have carefully reviewed and corrected the formatting errors on Page 17. Additionally, we have addressed formatting issues in other sections of the paper, including improving the presentation of formulas. Please see the revised manuscript for the corrected format.

Comments 4: Section 5: Conclusion need to be improved to highlight the objective of research work and demonstrate the importance of the paper to the field of ferroelectric nanofilms. Suggestions for further modification and improvement could be included.

Response: Noted. We have enhanced the conclusion to emphasize the research's importance in the ferroelectric nanofilm field. Suggestions for further modifications have been incorporated. Please review the updated conclusion in Section 5.

Comments 5: References- Most of the listed references are more than 5 years. Authors are encouraged to include newer and updated references.

Response: Thank you for your good suggestion. The following references are added for citation and reviewed in the introduction section:

  1. Wu, Y. B. Li, C. M. Zhou, H. Chen, S. W. Cheong, Y. W. Li, et al. Novel Geometric Ferroelectric EuInO3 Single Crystals with Topological Vortex Domains,Crystal Growth & Design, 2023, 23(3): 1980-1986.
  2. Wang, Y. Tang, Y. Zhu, X. Ma. Entangled polarizations in ferroelectrics: A focused review of polar topologies. Acta Materialia, 2023, 243:118485.
  3. Y. Wang, D. Liu, J. Wang, D. S. Liang, H. B. Huang. Phase-field simulations of topological conservation in multi-vortex induced by surface charge in BiFeO<sub>3</sub> thin films. Journal of Advanced Dielectrics, 2023.
  4. Mostovoy. Electrically-Excited Motion of Topological Defects in Multiferroic Materials. Journal of the Physical Society of Japan, 2023, 92(8).
  5. Liu, J. Wang, H. M. Jafri, X. Y. Wang, X. M. Shi, D. S. Liang, et al. Phase-field simulations of vortex chirality manipulation in ferroelectric thin films. Npj Quantum Materials, 2022, 7(1).
  6. X. Jiang, J. Wang. Stability of chiral polarization vortex in strained ferroelectric superlattices. Journal of Applied Physics, 2022, 131(16).
  7. Y. Huang, Q. A. Li, Y. Liang, X. Q. He, J. G. Wu, H. D. Fan, et al. Hierarchical-vortex polarization domain pattern in nano polycrystalline ferroelectric. Journal of Advanced Dielectrics, 2022, 12(04).
  8. Chen, C. B. Tan, Z. X. Jiang, P. Gao, Y. W. Sun, L. F. Wang, et al. Electrically driven motion, destruction, and chirality change of polar vortices in oxide superlattices. Science China-Physics Mechanics & Astronomy, 2022, 65(3).

Reviewer 2 Report

Comments and Suggestions for Authors

Yang et al. report in this manuscript, with title "Effect of width-to-length ratio of ferroelectric nanofilms on polarization vortex stability under uniaxial tension or compression", on PbTiO3 nanofilms with polarization vortex which are theoretically modeled, their vortex instability processes under uniaxial tension (or compression) being also simulated with the phase field theory. Overall, the work is well-investigated, the number of reported references is correct and the manuscript could be suitable to be published in the journal Materials.

Some points and revisions to be addressed:

1) In figures 1 and 2, a chart legend indicating the meaning of the reported parameters should be added.

2) The authors claim that their results are "very important for design of promising nano-ferroelectric devices based on polarization vortices", which should be a bit more developed in the Conclusions section. 

Comments on the Quality of English Language

In my opinion, and in general, minor editing of English language is required  

Author Response

Comments 1:In figures 1 and 2, a chart legend indicating the meaning of the reported parameters should be added.

Response: Thank you for your valuable suggestion. Chart legends have been added to Figures 1 and 2 to clarify the meaning of the reported parameters. Please refer to the revised manuscript for details.

Comments 2 :The authors claim that their results are "very important for design of promising nano-ferroelectric devices based on polarization vortices", which should be a bit more developed in the Conclusions section.

Response: Thank you for your comment. We acknowledge the importance of elaborating on the significance of our results for the design of nano-ferroelectric devices based on polarization vortices. We have made overall revisions to the conclusion section and added content related to application prospects, as follows:

In summary, adjusting the aspect ratio of nanofilms reveals crucial factors influencing the stability of polarization vortices, which holds significant implications for the de-sign and performance optimization of novel ferroelectric storage devices based on polarization vortices. Within the constraint of maintaining a constant nanofilm area, altering the aspect ratio to influence the shape of polarization vortices is of paramount importance for enhancing the performance of these innovative ferroelectric storage devices while pre-serving storage density.